# Searching for the Achilles’ Heel of Urethane Linkage—An Energetic Perspective

**DOI:** 10.3390/polym16081126

**Published:** 2024-04-17

**Authors:** Tamás Horváth, Karina Kecskés, Anikó Jordán Csábrádiné, Emma Szőri-Dorogházi, Béla Viskolcz, Milán Szőri

**Affiliations:** 1Institute of Chemistry, University of Miskolc, Miskolc-Egyetemváros A/2, H-3515 Miskolc, Hungary; kecskes.karina@student.uni-miskolc.hu (K.K.); aniko.jordan@uni-miskolc.hu (A.J.C.); emma.szori-doroghazi@uni-miskolc.hu (E.S.-D.); 2Higher Education and Industrial Cooperation Centre, University of Miskolc, H-3515 Miskolc, Hungary; bela.viskolcz@uni-miskolc.hu

**Keywords:** urethane bond, hydrogenation, hydrolysis, methanolysis, peroxidation, glycolysis, ammonolysis

## Abstract

A sudden increase in polyurethane (PU) production necessitates viable recycling methods for the waste generated. PU is one of the most important plastic materials with a wide range of applications; however, the stability of the urethane linkage is a major issue in chemical recycling. In this work, termination reactions of a model urethane molecule, namely methyl N-phenyl carbamate (MPCate), are investigated using G3MP2B3 composite quantum chemical method. Our main goal was to gain insights into the energetic profile of urethane bond termination and find an applicable chemical recycling method. Hydrogenation, hydrolysis, methanolysis, peroxidation, glycolysis, ammonolysis, reduction with methylamine and termination by dimethyl phosphite were explored in both gas and condensed phases. Out of these chemicals, degradation by H_2_, H_2_O_2_ and CH_3_NH_2_ revealed promising results with lower activation barriers and exergonic pathways, especially in water solvation. Implementing these effective PU recycling methods can also have significant economic benefits since the obtained products from the reactions are industrially relevant substances. For example, aniline and dimethyl carbonate could be reusable in polymer technologies serving as potential methods for circular economy. As further potential transformations, several ionizations of MPCate were also examined including electron capture and detachment, protonation/deprotonation and reaction with OH^−^. Alkaline digestion against the model urethane MPCate was found to be promising due to the relatively low activation energy. In an ideal case, the transformation of the urethane bond could be an enzymatic process; therefore, potential enzymes, such as lipoxygenase, were also considered for the catalysis of peroxidation, and lipases for methanolysis.

## 1. Introduction

Polyurethanes (PU), which account for approximately 8% of plastics [1], are highly versatile materials and hold a prominent position among the world’s most adaptable substances. In 2018, polyurethane secured the sixth position in global polymer production (20 million tons [2]), with a thriving market valued at 65.5 billion dollars. Predictions indicate that by 2025, this market is expected to soar to an impressive 105.2 billion dollars. Such a growing production of polyurethane necessitates the urgent need to discover viable recycling methods for the resulting waste to promote sustainable development.

Polyurethanes are produced from isocyanates and polyols with the aid of additive materials [3], such as carbon dioxide as a blowing agent, pentane to optimize insulation properties and various hydrocarbons as chain extenders. The composition of PU can be widespread allowing it to cover numerous fields. The most important part of polyurethane is the urethane bond, which is formed during the reaction between isocyanate and alcohol groups [1]. The stability of the urethane linkage is a major issue in the recycling of polyurethane-based plastics. Although landfill and incineration are still widely used as disposal method for all waste PU materials, physical, chemical and thermo-chemical recycling recovery gains are of increasing interest with large scale industrial applications already in operation [4].

The industry utilizes mechanical processes to recycle flexible polyurethane foam, specifically for the purpose of producing padding-type products. This process involves mixing the scrap foam with binders, which helps create the desired products. The adoption of this physical recycling method leads to what is known as the “carpet underlayment loop” [5]. Several chemical recycling technologies have been developed with different mechanisms of urethane bond degradation that have great potential. Glycolysis [6,7,8] and acidolysis [2,9] have been successfully used at the industrial scale, hydrolysis [10] has been introduced at the pilot scale, while aminolysis [11] and phosphorolysis [12] have been developed only at the laboratory scale.

Furthermore, catalytic hydrogenation could be a very effective tool for the processing of polyurethane waste, and it also offers an atom-economical approach to the deconstruction of PU [13]. Recently, Gausas et al. demonstrated [14] that the commercially available catalyst Ir-^iPr^MACHO, operating under 3 MPa H_2_ and temperatures ranging from 150 to 180 °C, serves as a versatile catalyst for effectively hydrogenating the four main types of polyurethane: flexible solid, flexible foamed, rigid solid, and rigid foamed. This process led to the isolation of aromatic amines and a polyol fraction. The use of isopropyl alcohol as a green solvent yielded the best results for the hydrogenation reactions. The success of this PU deconstruction method could be attributed to the combination of partial glycolysis and subsequent catalytic hydrogenation of PU fragments in solution, which could facilitate a cost-effective recycling of PU materials [14]. 

Johansen et al. [15] during their transition metal-catalyzed depolymerization studies, observed the ability of tertiary alcohols to deconstruct PU. The solvolysis test on 20 different PU materials yielded unaltered polyol and dianilines under mild reaction conditions. Solvolysis with tert-amyl alcohol (TAA) provides both the polyol and the dianiline precursor of the initial isocyanate without requiring expensive metal catalysts or other chemicals. They claim that the process could be scaled up for breakdown of greater amounts. TAA also has the advantage of being a low-boiling solvent making the separation from the reaction mixture much easier.

One of the earliest investigated reactions mainly for flexible polyurethane foam (PUF) waste recycling is hydrolysis, which yields polyol, amine intermediates, and carbon dioxide [1,3]. The process is conducted in an anaerobic environment and at high temperatures (above 150–320 °C). It is claimed that such chemical recycling of polyether polyol based PUFs is based on the cleavage of the urethane bonds, leaving the ether groups in polyether polyol intact [16]. The main disadvantage of hydrolysis is the high energy requirement to ensure adequate pressure and temperature [17]. Chaffin et al. studied the in vitro hydrolysis of polyether urethanes (PEU). They found that the urethane bonds in the backbone of PEU undergo chain scission when reacting with water, and this process is governed by an activation energy of approximately 90 kJ/mol at 37 °C [18]. 

Liu et al. [19] depolymerized thermoplastic polyurethane in sub- and supercritical methanol. During product analysis, it was determined that TPU (thermoplastic polyurethane elastomer) monomers and their methylates were formed. The temperature range of 220 °C to 240 °C was classified as the subcritical region, while temperatures exceeding 240 °C were considered the supercritical region. The degradation process and morphological changes of thermoplastic polyurethane (TPU) were clearly observed under different degradation conditions [19,20]. Methanol molecules, with their O- attaching to the C^+^ of -NHCOO-, caused the urethane bond to rupture through nucleophilic transesterification. Zamani et al. conducted a study on the thermal decomposition of 1,3-diphenyl urea, focusing on the formation of isocyanates. The reactions carried out in their investigation showed promising outcomes, resulting in the production of phenyl isocyanate and aniline [21]. 

Ammonolysis involves the use of ammonia in a depolymerization reaction, while aminolysis employs alkyl amines to produce polyols from polyurethane [11]. Olazabal et al. [22] studied the aminolysis of both aliphatic and aromatic polyurethanes and revealed high conversion rates when using different amines. The process involved employing the nucleophile agent (2-(methylamino)ethan-1-ol)) in excess, conducting the reactions under a nitrogen atmosphere, utilizing an organic base (triazabicyclodecene; TBD) and an organic acid (methanesulfonic acid; MSA) as catalysts. The study demonstrated that primary amines non-selectively cleave both the C-O and C-N bonds, resulting in the formation of an amine and a polyol. In contrast, secondary amines enable selective cleavage of the C-O bond, yielding diurea compounds with high yields. This selective cleavage process avoids the release of toxic amines and generates suitable monomers.

The reaction between the urethane group and the ester alkoxy group of phosphoric and phosphonic acids results in a mixture of phosphorus-containing oligouretanes with improved flame retardancy, although this method is not useful for circular economy.

PU biodegradation, including fungal, bacterial, and enzymatic degradation [1], could be a promising alternative, although the urethane bond is generally resistant to microbial influences [23]. Most so-called polyurethane-degrading enzymes only hydrolyze ester bonds and not the urethane bonds [23]. Ureases, on the other hand, can hydrolyze the urea bond in polyurethane, releasing two amines and carbon dioxide. Enzymes promoting polyurethane degradation must be optimized to withstand the production conditions [24]. Although there are studies on the enzymatic degradation of polyurethane, there is no enzyme or enzyme complex that can be used industrially for high yields of polyurethane degradation.

Previously, extensive theoretical and experimental investigations were conducted to comprehend the energetics and reaction mechanisms involved in the production of intermediates [25,26,27,28,29] and the formation of the urethane bond [30,31,32,33,34]. To complete this cycle, the objective of our present study was to identify potential attack points on the bonds of polyurethane by employing computational chemistry methods and reviewing the currently available recycling technologies. We also aimed to provide thermodynamical background to the understanding of the potential urethane termination reactions and give an upper limit for the barrier height using the uncatalyzed reactions illustrated in Figure 1.

## 2. Computational Methods

G3MP2B3 composite model [35] was applied to compute thermochemical properties of the species involved, such as zero-point corrected relative energy (∆E_0_, values are in the Appendix A), relative enthalpy (∆H, see Appendix A), standard enthalpy of formation (∆_f,298.15K_H(g)) and relative molar Gibbs free energy (∆G) as implemented in the Gaussian16 program package [36]. According to this protocol, geometry optimizations and frequency calculations are carried out using the B3LYP/6-31G(d) level of theory [37]. Scaling the harmonic wavenumbers obtained at B3LYP/6-31G(d) level of theory by a factor of 0.96 [35] is necessary to refine the accuracy of thermodynamic properties. Confirmation of transition state (TS) structures occurred through visual inspection of the intramolecular motions corresponding to the imaginary wavenumber using GaussView 6 [38]. Intrinsic reaction coordinate (IRC) calculations [39] were also carried out to map the minimal energy pathways (MEP). To refine the electronic energy, additional single point calculations of the critical points of the PES are implemented in this model, including QCISD(T)/6-31G(d) and MP2/GTMP2 levels of theories based on B3LYP/6-31G(d) geometries. To scan the solvent effect of water (ε_r_ = 78.4) and aniline (ε_r_ = 6.8) on the surrounding environment, the G3MP2B3 procedure via SMD polarizable continuum model developed by Truhlar et al. was used [40]. However, the accuracy of G3MP2B3 has already been demonstrated for similar reaction systems [25,28]. Accordingly, gas phase standard heat of formation values obtained here at G3MP2B3 were compared for further verification purposes. The gas phase standard heat of formation values at T = 298.15 K, ∆_f,298.15K_H(g), were achieved using an atomization scheme (AS) [41] and isodesmic reaction (IR). Accurate literature data necessary for AS calculation were collected from Computational Chemistry Comparison and Benchmark Database (CCBDB) [42], Ruscic’s Active Thermochemistry Tables [43], and Burcat’s Extended Third Millennium Ideal Gas and Condensed Phase Thermochemical Database for Combustion with Updates from Active Thermochemical Tables [44]. Thermochemical data for species not found in the above-mentioned databases were collected from other literature (cross referenced in the relevant tables) or calculated by group additivity (GA) role using the online NIST tool [45]. 

## 3. Results

### 3.1. Thermochemical Properties of the Reactants and Products

Thermochemical properties of all the reactants and products was calculated from G3MP2B3 results in the gas phase at thermodynamic standard conditions. To estimate standard molar enthalpy of formation (∆_f,298.15K_H^0^ (g)) values, the atomization scheme (AS) with accurate atomization enthalpy values (∆_atom_H^0^(^3^C) = 716.68 ± 0.45 kJ/mol, ∆_atom_H^0^(^2^H) = 218.00 ± 0.01 kJ/mol, ∆_atom_H^0^(^4^N) = 472.68 ± 0.40 kJ/mol, ∆_atom_H^0^(^3^O) = 249.23 ± 0.01 kJ/mol, ∆_atom_H^0^(^4^P) = 316.50 ± 1.00 kJ/mol) from the CCBDB database [42] and a hypothetical reaction with stochiometric numbers x, y, z, q and w were used:x^3^C + y^2^H + z^4^N + q^3^O + w^4^P = C_x_H_y_N_z_O_q_P_w_

The calculated ∆_f,298.15K_H^0^ (g) values were compared with values from literature (Table 1). For several chemicals and products, we did not find thermochemical data, so the NIST group additivity tool (GA) [45] was used to generate comparable data to the calculated values. The G3MP2B3 calculations are consistent with the literature data, the highest deviation is for methyl-hydrogencarbonate (17.5 kJ/mol). For the methyl phenyl urethane, only the crystalline phase enthalpy of formation had been found which is indicated in the table. Group additivity values for methyl phenylamine and (carboperoxyoxy)methane are also consistent, but in the case of methyl methylcarbamate, there is a higher deviation from the G3MP2B3 values, which is probably due to missing correction terms in the calculated group additivity data. In the case of methyl methylcarbamate, the group increment for the carbamate (urethane) group was calculated by subtracting the various incrementation values of the other groups from the enthalpy of formation of the methylcarbamate which was adopted from Burcat’s Database. Using this method, a group additivity value of −127 kJ/mol was obtained. Unfortunately, thermochemical data could not be generated in such way for the calculated phosphorus compounds.

### 3.2. Hydrogenation of the Polyurethane Molecule

To determine the energetics of recycling polyurethanes by hydrogenation, we investigated four reaction pathways (Figure 2) on our model urethane molecule. First, the hydrogenation of the amide (urethane) bond (Figure 2-TSa), then the saturation of the C-N bond between the amino and phenyl group (Figure 2-TSb), the H_2_-addition to the π-bond of the benzene ring (Figure 2-TSc) and lastly the hydrogenation of the C-O bond (Figure 2-TSd).

In addition to the gas-phase, calculations in aqueous and aniline solution were made; however, upon plotting the condensed phase data points (E**_aq_** and E**_aniline_**) as a function of the gas-phase results (E**_gas_**) a linear form can be observed (E**_aniline_** = (0.85202 ± 0.02292)∙E**_gas_** −(1.53095 ± 5.05515) for the aniline solution and E**_aq_** = (0.90446 ± 0.01486)∙E**_gas_** −(0.91806 ± 3.27754) for the aqueous solution) which makes the discussion of the solvent effect in higher detail less relevant. For this reason, only the gas phase results are discussed thoroughly in the text, although the reaction and activation Gibbs free energies of all three media are represented in the tables.

Hydrogenation of the urethane bond occurs in one step, in which the nucleophile nitrogen atom polarizes the hydrogen molecule, increasing the H-H bond length from 0.743 to 1.155 Å. In this process, the urethane bond length also increases by 0.22 Å while the O-H and N-H distances decrease forming aniline and methyl formate in the product complex. Standard Gibbs free energy of activation (Δ_TS;298.15K_G^0^) for the reactions are high (roughly 300 kJ/mol, Table 2). Overall, the reaction is slightly exergonic in water with Δ_R;298.15K_G^0^ = −6.4 kJ/mol (Table 2). Increasing the temperature to 100 °C did not affect the high energy barrier of the reaction, as well as the overall reaction free energies. It is important to highlight that both the reaction products, aniline and methyl formate, are important industrial materials. This would make the reaction relevant to the recyclization of urethanes. In the second pathway (via Figure 2-TSb) the electrophile carbon atom of the amino-phenyl bond coordinates the hydrogen while the C-N bond length increases by about 0.18 Å. Then the other hydrogen atom approaches the amino group transforming to a product complex consisting of benzene and methyl carbamate. Although the overall reaction is more exergonic, the activation energy values are nearly 100 kJ/mol higher than for the first reaction, making the hydrogenation of the phenyl and the amino group a secondary option compared to the mechanism of Figure 2-TSa. In the third reaction (Figure 2-TSc) hydrogenation of the π-bond on the benzene ring was investigated. First, the C-C bond of the ring increases by 0.1 Å, while the planar molecule distorts a slight amount. Then the hydrogens approach the carbon atoms reducing the aromacity of the molecule. In terms of energy, hydrogenation of aromatic compounds has a high energy barrier requiring harsh reaction conditions and catalysts. Our model urethane is no exception (Δ_TS;298.15K_G^0^ of 450 kJ/mol) making the reduction of the phenyl group infeasible.

Lastly, hydrogenation of the C-O bond (Figure 2-TSd) has been explored. The partially positive carbon (Mulliken charge of +0.602) atom polarizes the hydrogen molecule decreasing the C-H bond length to 1.592 Å. The oxygen atom of the bond then attracts the other hydrogen, resulting methanol and formanilide (N-phenyl formamide) in the process. This reaction has the second lowest energy barrier (Δ_TS;298.15K_G^0^ of 393.2 kJ/mol). All in all, the most favorable reaction pathway is the hydrogenation of the amide bond (Figure 2-TSa), which has the lowest energy barrier and is exergonic in water solution. A slight temperature dependence was observed. Due to the large difference in the activation free energy, one can hypothesize that Figure 2-TSa belongs to a well-separated hydrogenation channel without the production of significant amount of side-products.

### 3.3. Reaction Mechanism of Urethane Linkage Termination

As we concluded in the previous section, the C-N bond of the urethane linkage shows the highest reactivity which can be measured in the activation Gibbs free energy values in our polyurethane proxy. For this reason, the urethane termination mechanisms of the other chemicals were only explored in this position (Figure 3). In this study, the catalyst-free mechanism of the termination of the urethane linkage is investigated in one step, in which the nucleophile attacks the partially positive (Mulliken charges of +0.500) carbon atom in the linkage, while a proton transfer takes place forming aniline as one of the products. In the case of dimethylphosphite (Figure 3-TSk), instead of aniline, methyl-amine formation occurs due to the highly activated methyl group of the molecule. In terms of activation energies, it can be said that the overall Δ_TS;298.15K_G^0^ is very high which is in line with the high kinetic stability of polyurethanes. The most promising chemicals turned out to be hydrogen peroxide having the lowest energy barrier (171.9–199.2 kJ/mol, see Table 3), methyl amine and hydrogen which react with our model urethane in an exergonic pathway (−6.5 kJ/mol for hydrogenation and −25.3 kJ/mol for termination by methylamine in aqueous solution, Table 3). Increase of temperature to 100 °C from room temperature had a minimal effect on the high energy barriers, even increasing the Gibbs free energy of the reaction in some cases.

### 3.4. Reaction Mechanism of Urethane Linkage Termination by Water (Hydrolysis)

Hydrolysis of the urethane bond takes place in one step (Figure 3-TSe), where the nucleophile oxygen atom of the water molecule attacks the partially positive carbon atom of the urethane bond while the C-N bond length increases from 1.362 Å to 1.601 Å. In the process, a hydrogen atom is transferred between the water and the nitrogen atom resulting in aniline and methyl hydrogen carbonate molecules in the product complex. In terms of energies, the hydrolysis of the model urethane is endergonic with a high activation energy (Δ_TS;298.15K_G^0^ of 235.5 kJ/mol in gas phase); however, compared to the hydrogenation on the amide bond (TSa), there is a substantial reduction (ca. 100 kJ/mol) in the activation energy. 

### 3.5. Termination Mechanism of the Urethane Linkage by Methanol (Methanolysis)

Previous studies [20] suggested that the degradation of the urethane bond by methanolysis might be a viable reaction pathway. As seen from TSf of Figure 3, a methanol becomes a nucleophile after giving a proton to the nitrogen atom in the urethane molecule increasing the length of the C-N bond by 0.17 Å. Then the deprotonated methanol and the carbon atom of the urethane bond approach each other to form dimethyl carbonate as a product. The reaction is slightly endergonic, the lowest Δ_R;298.15K_G^0^ (5.7 kJ/mol) is in the gas phase; however, in terms of the activation free energy, a slight reduction has been observed in the condensed phase calculations compared to hydrolysis.

### 3.6. Reaction Mechanism of Urethane Linkage Termination by Hydrogen Peroxide (H_2_O_2_)

Hydrogen peroxide is a strong oxidizing agent and has various chemical applications. A study by Meijs et al. [49] reported that tensile properties of medical-grade polyurethanes can be decreased with the use of 25% (*w*/*w*) aqueous hydrogen peroxide at 100 °C. A similar reaction mechanism had been concluded in our investigation (Figure 3-TSg). A proton transfer takes place between the nucleophile nitrogen and the peroxide, while the carbon atom of the urethane bond attacks the deprotonated peroxide resulting in aniline and carboperoxyoxy methane as the products of the reaction. Termination by H_2_O_2_ has the lowest activation Gibbs free energy of all the explored reactions in aqueous solution at 298 K. The overall reaction is slightly endergonic making the termination thermodynamically controlled. Increase of temperature only mildly decreased the activation Gibbs free energy, while the reaction becomes slightly exergonic (−3.6 kJ/mol in Table 3) in aniline solution with the activation free energy of 172.3 kJ/mol (Table 3). 

### 3.7. Reaction Mechanism of Urethane Linkage Termination by Ethylene Glycol (EG)

H_2_O_2_ can be considered as two covalently bound hydroxyl groups which turn out to have advantage for urethane termination over methanol, water, and hydrogen molecule. Vicinal arrangement of two OH groups in ethylene glycol can be useful to measure the role of O-O over the number of OH group in the reactant. Even though previous studies have reported on the glycolysis of the ester bond, refs. [1,50] which helps in obtaining the components for polyurethanes called polyols, our research specifically investigated the process of glycolysis of the amide bond (Figure 3-TSh). First, a proton transfer takes place between the nucleophile, nitrogen, and one of the hydroxyl groups of the EG. Meanwhile the C-N bond length increases, and the deprotonated glycol attacks the carbon atom, producing 2-hydroxyethyl methyl carbonate and aniline in the product complex. The reaction is endergonic in both gas and condensed phases with the lowest Δ_R;298.15K_G^0^ in aqueous solution (16.7 kJ/mol). This reaction has significantly larger Δ_TS;298.15K_G^0^ compared to that of H_2_O_2_ (221.7 kJ/mol vs. 193.0 kJ/mol in gas phase). Increasing the temperature had a negative effect (activation energy raised to 228 kJ/mol) in the attempt to reduce the high energy barrier.

### 3.8. Reaction Mechanism of Urethane Linkage Termination by Ammonia (Ammonolysis)

The termination by ammonia (TSi in Figure 3) has a very similar reaction pathway to the hydrolysis. Ammonia is a good nucleophile due to the lone pair of electrons on the nitrogen atom in the molecule. This nucleophile attacks the carbon in the urethane bond decreasing the distance to 1.571 Å. In the same step, the bonding between one of the hydrogens from ammonia and the nitrogen of the urethane molecule result in aniline and methyl carbamate in the product complex. Overall, the reaction is endergonic with relatively low Δ_R;298.15K_G^0^ values but the reaction has even higher activation energies in the three media (gas phase, aniline and water solvation) compared to the hydrolysis.

### 3.9. Reaction Mechanism of Urethane Linkage Termination by Methyl Amine (MA)

Methyl amine is a derivative of ammonia where one of the hydrogen atoms is replaced by a methyl group, making it the simplest primary amine. It is considered a good nucleophile as it is highly basic (pK_b_ of 3.35 at 25 °C [51]). The reaction has a one-step mechanism like the other reactions. The nitrogen atom connected to the phenyl group takes a proton from the amino group while the C-N bond length of the urethane linkage increases from 1.363 to 1.975 Å. In the process, the deprotonated nucleophile attacks the carbon atom resulting in methyl methylcarbamate. The reaction has a nearly identical Gibbs free activation energy to the termination by methanol even at the higher temperature. However, the overall Gibbs free energy of the reactions are exergonic which makes the methyl amine a viable option in the research of polyurethane degradation.

### 3.10. Termination of the Urethane Linkage by Dimethyl Phosphite (DMP)

Studies reported by Troev et al. [52] showed promising results of polyurethane waste degradation by dimethyl phosphite without the use of any catalyst at a relatively mild temperature (142 °C) which is in line with the slight endergonicity of the reaction studied here. In terms of reaction mechanism, they have concluded that the carbon atom of the methoxy group of dimethyl phosphite acts as an electrophile attacking the nitrogen atom of the urethane bond. As a result, the length of the C-N bond increases (see TSk in Figure 3), while the negatively charged (Mulliken charge of −1.040) oxo group of the phosphite approaches the carbon atom of the amide bond resulting in methylphenylamine and MCMPA in the product complex. The results of the thermochemical calculations show that the reaction has the second highest activation energy. Increasing the temperature did not appear to be significant in terms of the activation energies or the overall reaction free energy.

### 3.11. Reaction Mechanism of Urethane Linkage Termination by Different Ionization Methods

Ionization of the urethane linkage can be a potential alternative of the molecular termination, considering that ionizing radiation of polymers causes crosslinking, scissoring or chain branching [53]. First, the gas-phase electron affinity (MPCate^−^) of the molecule was calculated (E_ea_ of 99.2 kJ/mol, in Appendix A, see Supporting Information) which corresponds to Δ_R;298.15K_G^0^ of 95.0 kJ/mol in Table 4. The latter quantity significantly reduced with the inclusion of solvents, especially water (a reduction of 211 kJ/mol can be observed). In terms of structural changes, it can be said that the planar molecule gets slightly distorted, and the amide bond length increases about 0.1 Å. As a second method, ionization energy of the urethane (MPCate^+^) had been examined. A high energy barrier can be observed with a high solvent effect (reduction in ionization energy of 195 kJ/mol in aqueous solution). Just like in the case of negative ionization, the amide bond length of the molecule increased; however, the negatively charged MPCate has a more favorable energy profile. Protonation of MPCate had been explored in two places of the molecule (attack at N, H^+^(1) and attack at O, H^+^(2) see Figure 4). The first path, the protonation of nitrogen, is endergonic (Δ_R;298.15K_G^0^ of 38.7 kJ/mol in the aqueous solution), while increasing the bond lengths of both the amide bond and the bond between the benzene ring and the nitrogen. The second pathway, the protonation of the oxo group has a higher reaction Gibbs free energy in water (152.3 kJ/mol) (Table 4). Increase of temperature had a minimal effect on the former protonation method, but protonation of the oxo group (H^+^(2)) reached a significant reduction in the reaction energies thus becoming more feasible in water. Bond length between the C-atom of the benzene ring and the nitrogen did not increase as much as in the previous method (1.446 Å compared to 1.497 Å of the first protonation method), while the amide bond length narrows down to 1.313 Å. 

Urethane termination by hydroxide anion had also been explored in two pathways, with a one-step mechanism and in both cases transition state structures were found ((OH^−^ (1)) and (OH^−^ (2)) of Figure 4). In the first mechanism, a negatively charged OH^−^ ion attacks the partially positive C-atom of the amide bond (Mulliken Charge +0.538) while giving its proton to the nucleophile nitrogen of the MPCate. In the second pathway, a proton transfer takes place between the OH^−^ ion and the nitrogen of the MPCate to form water and a negatively charged, deprotonated MPCate. Activation Gibbs free energy of the two reaction is very similar (Table 4), but the overall reaction Gibbs free energy is significantly lower for the second pathway. These values are even smaller than were found in the case of H_2_O_2_ reaction making the alkaline digestion an alternative for urethane termination.

### 3.12. Experimental Perspectives on Computed Energetics

Overall, based on our quantum chemical calculations the investigated depolymerization reactions of MPCate are associated with high activation energies, indicating that the termination of the urethane bond is energetically unfavorable. The lowest activation Gibbs free energy could be linked to the reaction with hydrogen peroxide; however, it still requires an energy input of nearly 200 kJ/mol (Table 3) suggesting that breaking the polymer chains of PU is a challenging task. Hydrogenation and aminolysis of MPCate also necessitate high activation energy which is in line with the conditions of the experiments in the literature [14,22]. Successful depolymerization with hydrogen required high reaction temperatures (150–180 °C) and long reaction times (approximately 20 h) even in the presence of Ir-iPrMACHO, a homogeneous, organometallic catalyst. This also implies that the urethane must be heated up first to depolymerize, requiring a considerable amount of energy investment. Similarly, a long reaction time (24 h) and high temperature reaction conditions (160 °C) were necessary for the depolymerization of another model polyurethane (IPDI-PU) in the experiment conducted by Olazabal et al. They achieved a good conversion using TBD:MSA (acid:base) catalyst under anaerobic conditions (nitrogen atmosphere) and a large excess of hexamethylenediamine.

All in all, we can conclude that polyurethane, thanks to its strong covalent bonds and stability, is highly resistant to chemical degradation and recycling. The high activation energies we have observed imply that considerable energy input is required to break these bonds and initiate depolymerization reactions. The endergonic nature of these reactions indicates that the energy balance is negative, further complicating the recycling process from an economic standpoint. In terms of an environmental standpoint, the best case scenario would be the substitution of PU for more eco-friendly materials.

Considering the results, the future for PU recycling might involve exploring alternative depolymerization techniques beyond traditional chemical methods, such as enzymatic or biological degradation, which could potentially offer more efficient and sustainable routes for PU recycling. 

### 3.13. Investigation of Reaction Catalysis by Enzymes 

In order to achieve environmentally friendly reactions, it is important to explore the potential of enzymes in the field of chemical reactions. Currently, microbiology has yet to discover enzymes that can effectively degrade polyurethane [54]. However, there is existing literature on how enzymes can break down ester bonds through hydrolysis, urea bonds through urease enzymes, or amide bonds through proteases [55]. While the microbiology industry is yet to find the key to biodegradation of polyurethane via enzymes, the industry already possesses recycling solutions for polyurethane through non-biological means [56]. However, reactions such as hydrogenation, hydrolysis, methanolysis, and peroxidation, are catalyzed by enzymes in biological environments. 

Hydrogenation plays a crucial role in two distinct processes: fatty acid biohydrogenation and CO_2_ hydrogenation to formic acid. In the context of fatty acid biohydrogenation, rumen bacteria are employed in the food industry to produce polyunsaturated fatty acids. This is a significant application that allows for the transformation of fatty acids via biohydrogenation [57]. On the other hand, CO_2_ hydrogenation represents a novel bio-industrial solution for hydrogen storage. This process involves the utilization of hydrogen-dependent CO_2_ reductase and format reductase enzymes. These enzymes enable the conversion of CO_2_ into formic acid, providing a promising avenue for hydrogen storage in a bio-based system [58]. Protein hydrolysis relies on serine proteases, such as trypsin. The catalytic triad found in trypsin shares similarities with the triad found in polyurethane-degrading enzymes (histidine, aspartate, serine) [59,60]. In the process of methanolysis, lipases, particularly the immobilized *Candida* sp. 99–125 lipase, play a crucial role. These lipases serve as catalysts for the production of biodiesel. They enable the conversion of lipids into biodiesel through methanolysis [61].

In this study promising outcomes are demonstrated by peroxidation. Hydrogen peroxide is a strong oxidizing agent which can degrade the urethane linkage. Peroxidation is a chemical reaction that primarily occurs between unsaturated fatty acids and reactive oxygen species, including the superoxide anion (O_2_^−^), hydrogen peroxide (H_2_O_2_), and hydroxyl radical (OH) [62]. This reaction leads to the oxidative damage of the fatty acids. Lipoxygenase, an enzyme of significant importance in biology, plays a pivotal role in lipid peroxidation by catalyzing the hydroperoxidation of lipids containing a cis,cis-1,4-pentadiene structure. This enzyme finds application in diverse fields, ranging from the oxidation of pigments in flour to the industrial production of dyes, coatings, detergents, polyvinyl chloride plasticizers, and even serving as an intermediate for drug synthesis [63]. We hope they can be also applied to the catalytic termination of urethanes in the near future.

## 4. Conclusions

The increase in polyurethane production requires a growing number of raw materials which also increases production costs and greenhouse gas emissions. Understanding the mechanism of urethane linkage termination is inevitable in search of a viable option to recycle our waste. The most eco-friendly and optimal scenario would be the application of biological degradation methods; however, the challenges associated with breaking down the urethane bond help us understand why it remains a significant hurdle, despite the advances in current technologies. On the other hand, chemolytic processes can be applied to break down PU waste recovering raw materials and precursors. After purification the reactants can be reused in production closing the circle of PU process. In our work, we studied chemical and ionization degradation methods from recent literature using quantum chemical methods, namely the G3MP2B3 composite model to develop more efficient and sustainable recycling technologies. Methyl phenylcarbamate (MPCate) was used as proxy for the calculations. Amongst the tested reactions, peroxidation, reduction by methylamine and hydrogenation of the amide bond proved to be most efficient, the former having the lowest activation Gibbs free energy of all the chemicals, while the amine reacted with the MPCate in an exergonic pathway. Hydrogenation might also be an encouraging option considering the product complex obtained consists of aniline and methyl formate, with exergonic behavior. Ionization had also been investigated using electron detachment/attachment, protonation of MPCate and reaction of MPCate with hydroxide anion. Reaction with OH^−^ has a relatively low Gibbs free activation enthalpy for both mechanisms, concluding that basic environment can be a significant degrading factor. 

Despite the high activation barriers, there is still hope for overcoming these obstacles through innovation in material design, research into catalysts and advancements in biodegradation. Achieving a circular PU economy will be a giant step forward for sustainable development. The goal of our study was to take action and understand the behavior of the urethane bond. Discovery of the most suitable recycling method lies in the future but with constant progress we get closer to the destination.

## Figures and Tables

**Figure 1 polymers-16-01126-f001:**
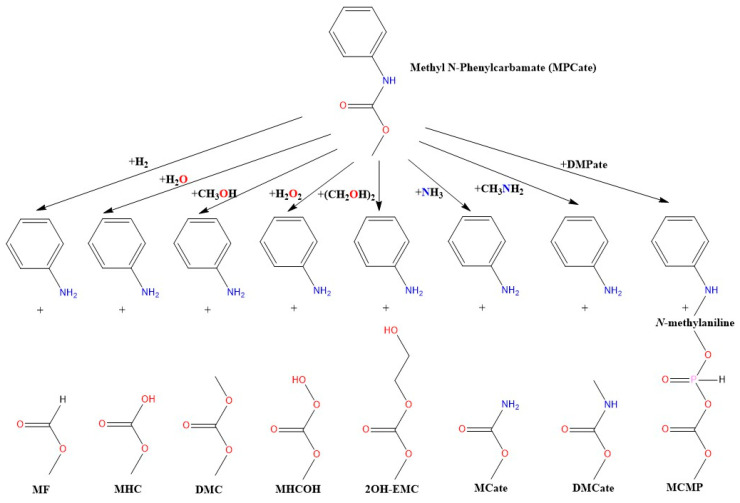
The studied molecular termination reactions of urethane bond using methyl N-phenylcarbamate (MPCate) as proxy. The abbreviation of the products are as follows: methyl formate (MF), methyl hydrogen carbonate (MHC), dimethyl carbonate (DMC), methyl hydrogen carbonoperoxoate (MHCOH), 2-hydroxyethyl methyl carbonate (2OH-EMC), methyl carbamate (MCate), dimethyl carbamate (DMCate), (methyl carbonic) (methyl phosphonic) anhydride (MCMP).

**Figure 2 polymers-16-01126-f002:**
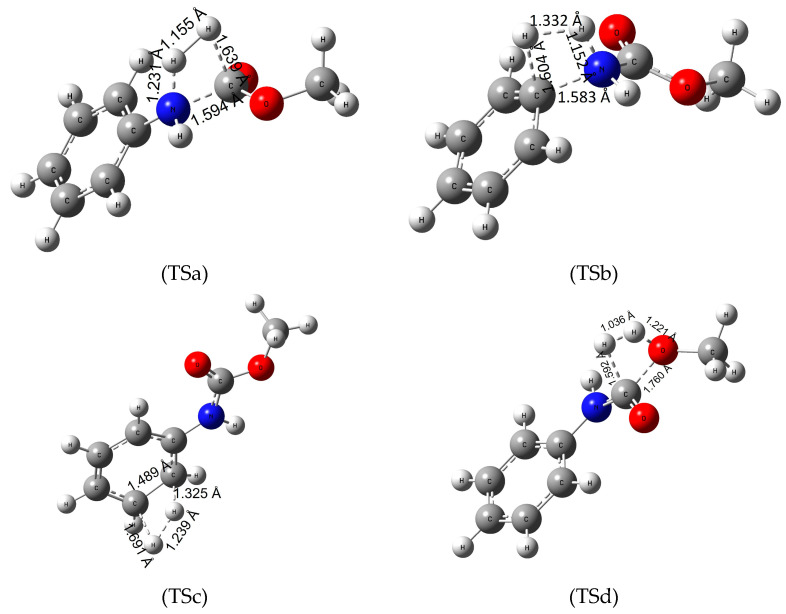
Transition state (TS) structures of the various hydrogenation pathways (TSa—hydrogeantion of the amide bond, TSb—hydrogenation of the C-N bond between the phenyl ring and the nitrogen, TSc—hydrogenation of the π-bond of the phenyl ring and TSd—hydrogenation of the C-O bond) of methyl N-phenylcarbamate (MPCate) as a proxy for the transformation of polyurethanes. The critical parameters (in Å) were obtained at B3LYP/6-31G(d) level of theory.

**Figure 3 polymers-16-01126-f003:**
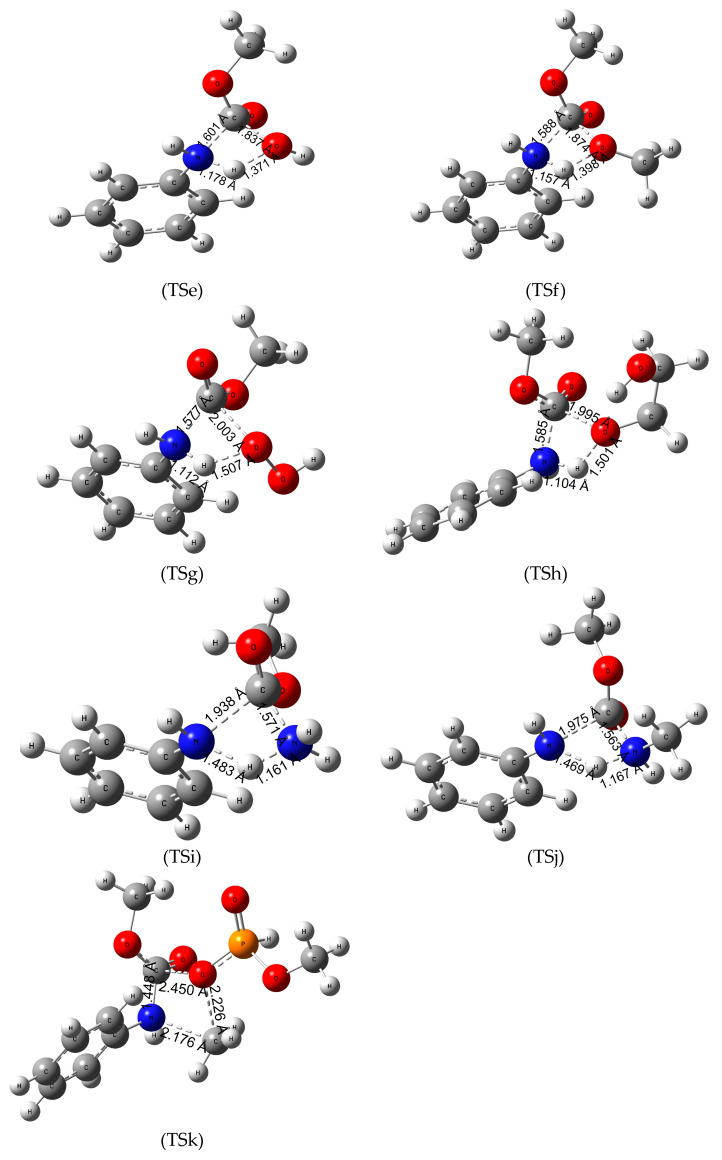
Transition state (TS) structures and the critical parameters (in Å) of termination reactions of MPCate by various chemicals TSe—hydrolysis, TSf—methanolysis, TSg—hydroperoxidation, TSh—glycolysis, TSi—ammonolysis, TSj—aminolysis, TSk—phosphorolysis obtained at B3LYP/6-31G(d) level of theory.

**Figure 4 polymers-16-01126-f004:**
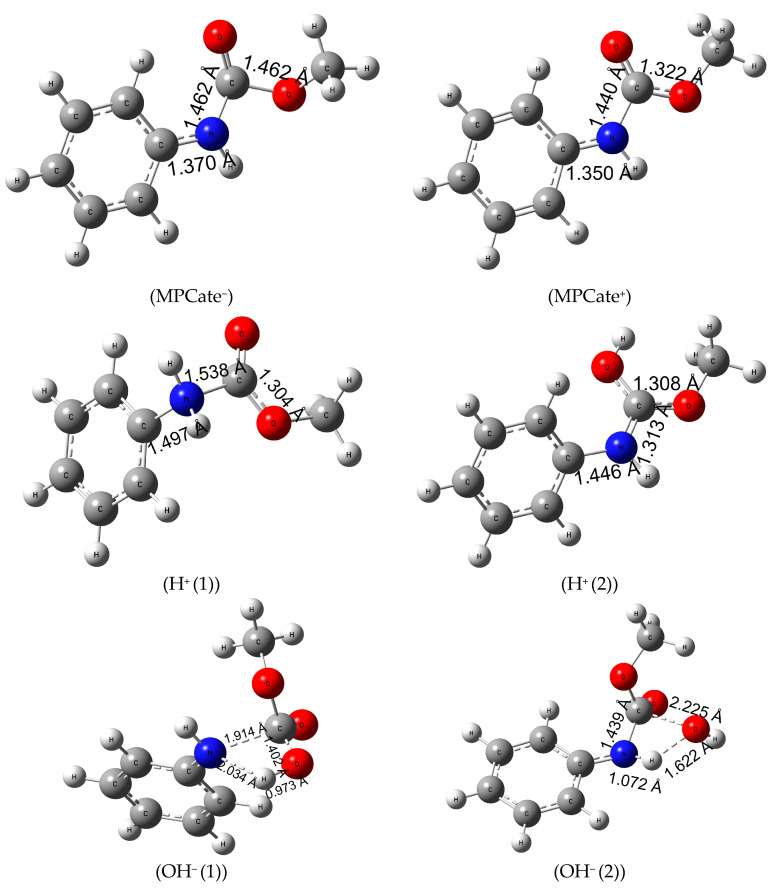
Structures and the critical parameters (in Å) obtained at B3LYP/6-31G(d) level of theory.

**Table 1 polymers-16-01126-t001:** Gas phase standard enthalpies of formation (∆_f,298.15K_H^0^ (g)) for reactants and products calculated from G3MP2B3 enthalpies (AS) compared to literature data.

Species	∆_f,298.15K_H^0^ (g) (kJ/mol)	Method	Ref.
(carboperoxyoxy)methane (C_2_H_4_O_4_)	−471.1	AS	
	−494.5	GA	
2-hydroxyethyl methyl carbonate (C_4_H_8_O_4_)	−752.9	AS	
	−753.5	GA	
ammonia (NH_3_)	−42.4	AS	
	−45.6	Burcat	
aniline (C_6_H_7_N)	85.6	AS	
	87.0	Burcat	
dimethyl carbonate (C_3_H_6_O_3_)	−569.3	AS	[46]
	−571.0	lit.	
dimethyl phosphite (C_2_H_7_O_3_P)	−758.3	AS	
ethylene-glycol (C_2_H_6_O_2_)	−387.9	AS	
	−389.4	Burcat	
formanilide (C_7_H_7_NO)	−64.8	AS	
	−55.2	GA	
H_2_O	−240.5	AS	
	−241.8	Burcat	
hydrogen-peroxide (H_2_O_2_)	−131.0	AS	
	−135.9	Burcat	
methanol (CH_3_OH)	−199.4	AS	
	−200.9	Burcat	
methyl carbonic methyl phosphonic anhydride (MCMPA-C_3_H_7_O_5_P)	−1113.6	AS	
methyl methylcarbamate (C_3_H_7_NO_2_)	−402.7	AS	
	−367.7	GA	
methyl phenyl urethane (C_8_H_9_NO_2_)	−286.4	AS	[47]
	−186.7 (cr)	lit.	
methylamine (CH_3_NH_2_)	−17.5	AS	
	−20.9	Burcat	
methylcarbamate (C_2_H_5_NO_2_)	−406.3	GA	
	−412.0	Burcat	
methylformate (C_2_H_4_O_2_)	−358.8	AS	
	−360.0	Ruisic	
methyl-hydrogencarbonate (C_2_H_4_O_3_)	−589.6	AS	[48]
	−607	lit.	
methyl-phenylamine (C_7_H_9_N)	92.4	AS	
	86.6	GA	
OH^−^	−138.5	AS	
	−139.0	Ruisic	

**Table 2 polymers-16-01126-t002:** Gibbs free energy values (in kJ/mol) for the termination of urethane linkage by hydrogen for different pathways in gas phase as well as in aniline and aqueous phases.

Reactant	Δ_R;298.15K_G^0^ (kJ/mol) [Δ_TS;298.15K_G^0^ (kJ/mol)]			Δ_R;398.15K_G (kJ/mol) [Δ_TS;398.15K_G (kJ/mol)]		
	Gas Phase (ε = 1)	Aniline (ε = 6.8)	Water (ε = 78.4)	Gas Phase (ε = 1)	Aniline (ε = 6.8)	Water (ε = 78.4)
H_2_ (TSa)	5.2 [339.6]	5.2 [311.0]	−6.4 [290.4]	7.3 [325.0]	0.9 [298.8]	−6.5 [281.6]
H_2_ (TSb)	−48.9 [424.9]	−56.0 [378.1]	−63.8 [362.0]	−47.7 [431.7]	−56.9 [384.8]	−64.8 [367.7]
H_2_ (TSc)	41.2 [452.2]	35.0 [373.9]	28.3 [329.1]	48.8 [459.4]	41.5 [379.6]	33.8 [334.3]
H_2_ (TSd)	57.9 [393.2]	40.9 [382.2]	35.5 [370.7]	5.4 [346.1]	−11.7 [335.1]	−15.5 [323.4]

**Table 3 polymers-16-01126-t003:** Gibbs free energy values (in kJ/mol) for the termination of urethane linkage by different reactants in gas phase as well as in aniline and aqueous phases.

Reactant	Δ_R;298.15K_G^0^ (kJ/mol) [Δ_TS;298.15K_G^0^ (kJ/mol)]			Δ_R;398.15K_G^0^ (kJ/mol) [Δ_TS;398.15K_G^0^ (kJ/mol)]		
	Gas Phase (ε = 1)	Aniline (ε = 6.8)	Water (ε = 78.4)	Gas Phase (ε = 1)	Aniline (ε = 6.8)	Water (ε = 78.4)
H_2_ (TSa)	5.2 [339.6]	5.2 [311.0]	−6.4 [290.4]	7.3 [325.0]	0.9 [298.8]	−6.5 [281.6]
H_2_O (TSe)	2.8 [235.5]	2.8 [220.5]	8.5 [205.7]	3.3 [239.3]	8.4 [224.9]	9.0 [209.9]
CH_3_OH (TSf)	5.7 [219.3]	9.8 [205.5]	11.1 [192.4]	4.5 [223.6]	8.9 [210.6]	9.5 [196.9]
H_2_O_2_ (TSg)	1.4 [193.0]	1.4 [171.9]	13.4 [164.1]	3.2 [199.2]	−3.6 [172.3]	13.2 [169.0]
C_2_H_6_O_2_ (TSh)	21.2 [221.7]	21.2 [204.5]	16.7 [192.8]	20.9 [228.2]	23.3 [209.7]	14.7 [197.2]
NH_3_ (TSi)	3.3 [250.2]	1.2 [225.5]	1.4 [216.9]	3.9 [254.8]	1.0 [230.0]	2.7 [222.0]
CH_3_NH_2_ (TSj)	−12.1 [229.7]	−14.1 [204.2]	−22.8 [193.1]	−14.8 [233.6]	−15.8 [208.4]	−25.3 [196.9]
C_2_H_7_O_3_P (TSk)	16.5 [318.0]	13.5 [291.0]	22.4 [277.4]	17.3 [323.3]	12.2 [292.2]	23.8 [280.8]

**Table 4 polymers-16-01126-t004:** Gibbs free energy values (in kJ/mol) for the termination of urethane linkage by different ionization methods in gas phase as well as in aniline and aqueous phases.

Reactant	Δ_R;298.15K_G^0^ (kJ/mol) [Δ_TS;298.15K_G^0^ (kJ/mol)]			Δ_R;398.15K_G^0^ (kJ/mol) [Δ_TS;398.15K_G^0^ (kJ/mol)]		
	Gas Phase (ε = 1)	Aniline (ε = 6.8)	Water (ε = 78.4)	Gas Phase (ε = 1)	Aniline (ε = 6.8)	Water (ε = 78.4)
MPCate^−^	95.0	−71.5	−116.3	92.9	−72.5	−117.5
MPCate^+^	789.4	616.9	594.5	788.8	616.7	594.5
H^+^ (1)	−833.1	67.5	38.7	−835.1	66.3	37.2
H^+^ (2)	−943.0	138.2	152.3	−849.5	66.6	36.2
OH^−^ (1)	108.5 [140.3]	89.3 [144.7]	65.9 [139.7]	112.4 [143.9]	94.0 [150.3]	69.4 [143.1]
OH^−^ (2)	16.0 [176.0]	17.7 [148.8]	6.7 [114.8]	16.8 [180.1]	19.2 [153.9]	6.7 [118.2]

## Data Availability

The data presented in this study are available on request from the corresponding author. The data are not publicly available due to the policy of the University of Miskolc.

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
