# Peer review of "Searching for the Achilles’ Heel of Urethane Linkage—An Energetic Perspective"

_polymers, 2024, doi:10.3390/polym16081126_

Round 1

Reviewer 1 Report

Comments and Suggestions for Authors

This manuscript appears to have some issues on referencing system.

Page 2, lin 54, [6], [7], [8] => [6-8]

Page 3, line 117, [25], [26], [27], [28], [29] => [25-29]

Page 9, line 296, , (Hiba! A hivatkozási forrás nem található.), …???

Page 12, line 385, (Hiba! A hivatkozási forrás nem található.), …???

……

Polyurethane material is not composed of urethane bonds only. It contains catalyst, surfactant, dispersant, blowing agent, flame retardant, plasticizer, UV yellowing agent, impact filler, etc.

When authors claim that peroxidation and reduction by methylamine show promising results, it needs to be verified. This G3MP2B3 composite model should be a good tool if it is verified with experimental data.

This computational chemistry of polyurethane depolymerization reactions seems fine. However, it needs to be compared with the experimental data, at least, a few of them.

Reviewer 2 Report

Comments and Suggestions for Authors

Abstract

1.     Page 1: The abstract should be more meaningful and contain the most important findings and results of the paper; rewrite it.

Keywords

2.     Page 1, Line 30: Delete ab initio from keywords.

 Introduction

3.     Revise the Introduction part and mention a few examples from recent literature research carried out in this field and previous advancements in the current field.

4.     Page 2, Line 48, 53: Merge the two paragraphs into one paragraph.

5.     Page 2,3, Line 88, 103: Merge the two lines with the above paragraph. Do not make two lines as a paragraph.

Page 4:  Results

6.     As mentioned in Fig 2, what does TSa, b c, d mean? In the caption, elaborate each individual figure using Roman numbers (i), (ii), etc.  

7.     Page 7, Line 240. Table 2; why TSb shows negative Gibbs free energy values compared to TSa, TSc, and TSd. 48.9 [424.9] -56.0 [378.1] -63.8 [362.0] -47.7 [431.7] -56.9 [384.8] -64.8 [367.7]

8.     Page 8, Line 244. Correct the Figure caption.

References

10.  Page 15: The author should check all the formats of the references.

11.  Page 18, Line 587: There is a major issue in references; the style is inaccurate, i.e., Ref [42], [44], [43]. There is no consistency, and references aren’t up to date. 

Round 2

Reviewer 1 Report

Comments and Suggestions for Authors

This G3MP2B3 composite model should be a good tool if it is verified with experimental data.

This computational chemistry of polyurethane depolymerization reactions would be fine. However, it needs to be compared with the experimental data, at least, a few of them.

Comments on the Quality of English Language

Overall it is good but there were some spelling errors.

For example, on Page 15, line 447, precursors.

Author Response

Dear reviewer, 

Reviewer 2 Report

Comments and Suggestions for Authors

Accept in present form.

Author Response

Dear reviewer,
